# Evaluation of the Effect of Influenza Vaccine on the Development of Symptoms in SARS-CoV-2 Infection and Outcome in Patients Hospitalized due to COVID-19

**DOI:** 10.3390/vaccines12070765

**Published:** 2024-07-12

**Authors:** Jose Roberto Gutierrez-Camacho, Lorena Avila-Carrasco, Araceli Gamón-Madrid, Jose Ramon Muñoz-Torres, Alberto Murillo-Ruiz-Esparza, Idalia Garza-Veloz, Perla M. Trejo-Ortiz, Fabiana E. Mollinedo-Montaño, Roxana Araujo-Espino, Iram P. Rodriguez-Sanchez, Ivan Delgado-Enciso, Margarita L. Martinez-Fierro

**Affiliations:** 1Doctorado en Ciencias con Orientación en Medicina Molecular, Unidad Academica de Medicina Humana y Ciencias de la Salud, Universidad Autonoma de Zacatecas, Zacatecas 98160, Mexico; rob_gutierrez_mm@uaz.edu.mx (J.R.G.-C.); araceli.gamon@uaz.edu.mx (A.G.-M.); jose.rmt26@gmail.com (J.R.M.-T.); idaliagv@uaz.edu.mx (I.G.-V.); perlatrejo@uaz.edu.mx (P.M.T.-O.); fabiana.mollinedo@uaz.edu.mx (F.E.M.-M.); roxana.araujo@uaz.edu.mx (R.A.-E.); 2Instituto Mexicano del Seguro Social, Hospital General de Zona No. 1, Zacatecas, Zacatecas 98000, Mexico; albertomre05@hotmail.com; 3Laboratorio de Fisiologia Molecular y Estructural, Facultad de Ciencias Biologicas, Universidad Autonoma de Nuevo Leon, San Nicolas de Los Garza 66450, Mexico; iramrodriguez@gmail.com; 4Department of Molecular Medicine, School of Medicine, Cancerology State Institute, IMSS-Bienestar, University of Colima, Colima 28040, Mexico; ivan_delgado_enciso@ucol.mx

**Keywords:** influenza vaccine, SARS-CoV-2, COVID-19, hospitalization

## Abstract

Background: COVID-19 is an infectious disease caused by SARS-CoV-2. It is unclear whether influenza vaccination reduces the severity of disease symptoms. Previous studies have suggested a beneficial effect of influenza vaccination on the severity of COVID-19. The aim of this study was to evaluate the possible protective effect of the influenza vaccine on the occurrence of SARS-CoV-2 infection symptoms and prognosis in patients hospitalized with COVID-19. Methods: This was a retrospective cohort study of patients who tested positive for SARS-CoV-2, identified by quantitative real-time polymerase chain reaction. Chi-square tests, Kaplan–Meier analysis, and multivariate analysis were performed to assess the association between influenza vaccination and the presence of symptoms in hospitalized patients with COVID-19 and their outcome. Results: In this study, 1712 patients received positive laboratory tests for SARS-CoV-2; influenza vaccination was a protective factor against the presence of characteristic COVID-19 symptoms such as polypnea, anosmia, dysgeusia, and fever (*p* < 0.001). Influenza-vaccinated patients had fewer days of hospitalization (*p* = 0.029). Conclusions: The findings of this study support that influenza vaccination is associated with a decrease in the number of symptoms in patients hospitalized due to COVID-19, with fewer days of hospitalization, but not with the outcome of disease.

## 1. Introduction

Severe acute respiratory syndrome coronavirus 2 (SARS-CoV-2) caused the 2019 coronavirus disease (COVID-19) pandemic, leading to high morbidity and mortality rates worldwide [1,2]. So far, it has resulted in 770,875,433 cases of infection and 6,959,316 deaths, with significant global health and economic consequences [3].

On the other hand, seasonal influenza occurs annually from fall through spring, and it is usually caused by influenza A or B viruses. This infection has been associated with an increase in mortality and morbidity if there is SARS-CoV-2 co-infection [4,5,6,7]. Annual influenza vaccination is recommended by the World Health Organization (WHO) to prevent infections and severe complications, especially in high-risk groups such as pregnant women and older adults [8,9].

Due to the similar transmission mechanisms, researchers have looked at the relationship between SARS-CoV-2 infection and influenza immunity [10,11,12]. At present, there seems to be no compelling evidence of a negative impact of influenza vaccination on COVID-19 patients [13]. While some authors have shown that the influenza vaccine is not linked to incidence [14,15], severity [15,16], and mortality [15,17] of COVID-19, others have found some associations [18,19,20].

The fact that the use of influenza vaccines is widespread points to the need to clarify whether these vaccines may affect the outcome of SARS-CoV-2 infections. Actually, certain vaccines have protective effects on innate immunity, a process termed trained immunity (Figure 1); therefore, a beneficial effect of influenza vaccination on susceptibility to COVID-19 is possible [15,21,22,23].

The clinical features of COVID-19 are broad, and its severity ranges from mild symptoms to acute respiratory distress syndrome and death [24,25,26,27,28]. Pre-existing liver, cerebrovascular, renal, and gastrointestinal diseases, hypertension, type 2 diabetes mellitus (T2DM), and chronic obstructive pulmonary disease (COPD) increase susceptibility to SARS-CoV-2 infection, and are associated with increased risk of mortality [1,29].

It is of utmost importance to know that patients with pre-existing cardiovascular risk factors have a high rate of probability of developing severe disease as a consequence of cardiovascular complications of COVID-19 [30,31,32,33,34].

Humanity finds itself in a time of SARS-CoV-2 disease (COVID-19), which places an additional burden on healthcare systems, and creates the potential for co-infection of SARS-CoV-2 and influenza [35,36,37,38]. 

It has been suggested that receipt of influenza vaccination results in significant protection against most influenza viruses [11]. These are of greatest concern in the elderly and in patients with pre-existing medical conditions which could cause them to experience increased mortality and morbidity with influenza infection [39], in addition to being at increased risk of severe COVID-19 infection [1,29]. Influenza has already been reported to be associated with an increased risk of aortic dissection [40], myocarditis [41], myocardial infarction, stroke, and death in persons with cardiovascular disease [42,43,44]; it has also already been demonstrated that the administration of influenza vaccine reduces the risk of major adverse cardiovascular events in this population [45,46,47,48]. Therefore, year-to-year vaccination against influenza has been identified as an outstanding intervention to reduce the risk of cardiovascular events in patients with atherosclerotic or coronary vascular disease [49]. The increased risk of cardiovascular complications due to COVID-19 in these patients, in concert with the possible protective effect it generates against COVID-19, further supports the importance of influenza vaccination in this population [30,31,32,33].

The potential protection of influenza vaccination against the severity of SARS-CoV-2 infection, attributed to the induction of cross-neutralizing antibodies and T-cells [10,17,50,51], becomes relevant with the onset of the influenza season. In the light of the aforementioned, this study aimed at assessing the potential protective effect of influenza vaccine on the occurrence of SARS-CoV-2 infection symptomology, and on the outcome of patients hospitalized due to COVID-19. 

## 2. Materials and Methods

### 2.1. Study Design

This cross-sectional study included hospitalized adult patients with a confirmed diagnosis of COVID-19 registered at Hospital General de Zacatecas Luz González Cosío (Zacatecas, Mexico) between 10 January 2020 and 31 December 2020. This study was revised and approved by the Ethics and Research Committee at the Zacatecas General Hospital Luz González Cosío (ID 02023/2019).

Study population: Laboratory-confirmed SARS-CoV-2 infection was diagnosed by real-time reverse transcriptase–polymerase chain reaction (RT-PCR) assay using a nasal/pharyngeal specimen during or prior to hospital admission. Sociodemographic and detailed clinical data of the patients, including comorbidities and history of influenza immunization, were recorded. Hospitalized patients with documented “do not intubate and/or do not resuscitate” orders and those who left the hospital against medical advice were excluded from the study. Patients who did not meet clinical endpoints for intubation were also excluded. Clinical outcomes, such as discharge, mortality, intubation, and comorbidities, were also obtained.

It is important to note that the patients who in this study were considered as vaccinated against influenza were the patients who received the vaccine in the 2019–2020 National Anti-influenza Vaccination Campaigns that began on 16 October 2019, which consisted of the application of one dose of trivalent anti-influenza vaccine type A and B [52]. 

### 2.2. Sample Size Calculation

The size of sample calculated for the study was 1712 patients, and it was obtained online (https://www.questionpro.com/sample-size-calculator/, accessed on 3 September 2022), using the formula proposed by Murray and Larry (2005) [53] with a confidence level of 95%, *p* = 0.5, a margin of error of ±2.3679%, and the population of the State of Zacatecas, which is 1,622,138 inhabitants.

### 2.3. Data Analysis

Descriptive statistics and chi-square tests were used to compare clinicopathological characteristics of patients by influenza vaccination status. Odds ratios (ORs) with 95% confidence intervals (CIs) were computed for the association between dichotomous influenza immunization and the outcomes under study (i.e., symptomatology and mortality). Subsequently, the Kaplan–Meier method was used to plot the survival curve according to influenza vaccination status and days of hospitalization by COVID-19. Multivariate logistic regression was also performed to identify the association between comorbidities, risk factors, and the role of influenza vaccination on mortality in patients hospitalized with COVID-19. Statistical analysis was performed in SPSS software (version 13.0, SPSS Inc., Chicago, IL, USA). *p*-values < 0.05 were considered statistically significant.

## 3. Results

In this study, 1712 patients were included. The age of the patients ranged from 20 to 86 years, with a mean age of 63.7 years 1–2; 978 (57.1%) were men and 734 (42.9%) were women. Almost 30% (29.7%) had received the influenza vaccine (trivalent influenza vaccine A and B), mostly men (60.7%). Of the 1203 patients without the influenza vaccine, 669 (55.6%) were male.

Symptoms reported by vaccinated patients included cough (100%), attack to the general state (89.9%), headache (83.4%), thoracic pain (77.9%), and fever (66.6%). Decreased ORs were seen in vaccinated patients for rhinorrhea (OR = 0.6; 95% CI: 0.4–0.8, *p* = 0.001), polypnea (OR = 0.4; 95% CI: 0.3–0.6, *p* < 0.001), cyanosis (OR = 0.6; 95% CI: 0.4–0.8, *p* < 0.001), anosmia (OR = 0.5; 95% CI: 0.4–0.7, *p* < 0.001), dysgeusia (OR = 0.4; 95% CI: 0.3–0.5, *p* < 0.001), fever (OR = 0.5; 95% CI: 0.4–0.7, *p* < 0.001), and arthralgias (OR = 0.6; 95% CI: 0.5–0.8, *p* < 0.001).

Respiratory symptoms such as cough, thoracic pain, and general symptoms, such as headache and chills, were present in a greater proportion of patients vaccinated when they were compared with non-influenza-vaccinated patients (*p* < 0.05; OR range: 1.1 to 2.4). Comorbidities such as T2DM and hypertension represented a high percentage in both the vaccinated (38.7% and 55.5%, respectively) and non-vaccinated (49.3% and 62.9%, respectively). Hypertension showed less risk of being present in patients vaccinated against influenza, and was 0.7 (95% CI: 0.5–0.9, *p* = 0.005) (Table 1).

For the survival analysis of patients hospitalized with COVID-19, both the data of patients vaccinated against influenza and those not vaccinated were considered, as well as the days spent in hospital. There was an average of 3.7 days of hospitalization in vaccinated patients versus 6.9 days of hospitalization in non-vaccinated patients, obtaining a statistically significant difference (*p* = 0.029) in the Long Rank test (Mantel–Cox) (Figure 2).

Using multivariate analysis, we observed that there was no association between the outcome of patients with COVID-19 and influenza vaccination status (OR = 0.9; 95% CI: 0.6–1.2, *p* = 0.657); however, critical condition, such as intubation, was associated with death in patients with COVID-19 (OR = 12.8; 95% CI: 7.0–20.5, *p* < 0.001), as well as other comorbidities such as T2DM (OR = 2.3; 95% CI: 1.7–3, *p* < 0.001), heart disease (OR = 5.1; 95% CI: 2.6–9.7], *p* < 0.001), obesity (OR = 9.5; 95% CI: 7–12.9, *p* < 0.001), and tobaccoism (OR = 2.4; 95% CI: 1.6–3.8, *p* < 0.001) (Table 2).

## 4. Discussion

Influenza vaccination is considered the most effective measure to prevent influenza and influenza-related complications, especially in high-risk populations. It has been reported that the mortality rate for COVID-19 appears to be higher than for seasonal influenza, with the caveat that both diseases primarily affect older adults (>65 years). This higher mortality rate from COVID-19 could be due to complications that develop depending on underlying comorbidities of the patients, population immunity, viral pathogenicity, and the immune system response of the patient. The aim of this study was to evaluate the possible protective effect of influenza vaccine on the occurrence of SARS-CoV-2 infection symptoms and the outcome in patients hospitalized due to COVID-19. According to our results, the application of the influenza vaccine influences a decrease in the presence of symptoms such as fever, diarrhea, myalgia, arthralgia, rhinorrhea, polypnea, cyanosis, and anosmia in patients infected with SARS-CoV-2. Other symptoms such as cough, chest pain, colds, headache, and general condition attack were more likely to occur in vaccinated patients compared to unvaccinated subjects; these results are in agreement with previous reports [54,55]. 

A retrospective cohort study in Michigan, USA, describes that patients who received an influenza vaccine tended to have more comorbidities than those in the unvaccinated group, including higher rates of chronic lung disease, congestive heart failure, T2DM, and high blood pressure. Compared to this study, we observed that patients with comorbidities such as T2DM, hypertension, heart disease, and COPD were less likely to be hospitalized for COVID-19 if they had received influenza vaccination compared to non-vaccinated patients. This could be explained by the fact that patients with pre-existing medical conditions often receive preventive therapies such as influenza vaccination [56].

It has been suggested that when analyzing the impact of influenza vaccination with COVID-19 risk factors for complications, they observed a reduced risk in patients with cardiovascular disease, T2DM, pulmonary disease, and chronic kidney disease [57]. In comparison, in the present report, we observed a decreased rate and risk of hospitalization in patients with T2DM, COPD, hypertension, and heart disease who were vaccinated against influenza. It was published in 2020 that patients with COVID-19 who received an inactivated trivalent influenza vaccine were more likely to survive and required less intensive hospital care than patients without influenza vaccination [10]. Data from this research suggest that patients vaccinated against influenza are more likely to leave the hospital due to improvement compared to patients who were not vaccinated against influenza. Additionally, Ragni et al. (2020) described a lower risk of hospitalization and death in patients with COVID-19 who had received influenza vaccine [51]. 

One possible explanation for the protective effect of influenza vaccination in some of the patients who did receive the dose is that unvaccinated individuals are at risk of persistent viral infections, which may lead to a decrease in T-cell counts that, in turn, affects the immune response to SARS-CoV-2 [58,59]. It is also known that influenza vaccination does not induce a strong virus-specific CD8 T-cell immune response compared with natural infection, and although this comparison may be a disadvantage of inactivated influenza vaccines, it is beneficial for eliminating SARS-CoV-2 infection, as vaccinated persons will have more T-cell diversity compared with persons who were naturally infected with influenza [60,61].

It is noteworthy that influenza virus alters the cytotoxic effect of natural killer (NK) cells and induces apoptosis, which ultimately alters host immune defense mechanisms against SARS-CoV-2 in the acute phase of the disease [62,63,64]. Patients who did not receive influenza vaccination have a higher proportion of influenza-specific resident memory T cells (T_RM_), which are highly proliferative and highly productive of inflammatory cytokines in the lungs [65,66], which is associated with an exaggerated severe ARDS inflammatory response seen in some patients with COVID-19 [67]. This immunological mechanism that explains the protective effects of influenza vaccine against COVID-19 is a process called trained immunity [68].

This immune response to viral pathogens is a fundamental feature of the adaptive immune system, leading to more rapid immune responses after subsequent reinfection [69]; these responses are most frequently directed to viral surface antigens, e.g., influenza hemagglutinin [70] and the SARS-CoV-2 spike protein [71]. Each antigen contains binding proteins for B- and T-cell receptors, termed epitopes [72]. Antibody responses to viral pathogens depend on B cells that “neutralize” the infection; most antibody responses are neutralizing antibodies, so their balance determines the efficacy of the response [73].

Influenza vaccine-induced trained immunity against SARS-CoV-2 induces the cytokine response at the onset of infection that is crucial for decreasing viral load and preventing systemic inflammation. The amplified IL-6 response activates acute-phase proteins, stimulates the development of effector T cells in conjunction with antibody production, forming the link between adaptive and innate immunity, leading to clearance of infection [74]. In contrast, anti-inflammatory cytokines such as IL-1Ra are necessary to counteract excessive inflammation; an increase in IL-6 production has been observed in parallel with IL-1Ra after stimulation with influenza vaccine, which contributes to maintaining a balance in the inflammatory state of patients [75]. It is also known that trained immunity is induced in NK cells, which suppress viral infections, through the production of IFN-γ, this production being after stimulation with influenza vaccine. There is a functional reprogramming of NK cells, which subsequently activate macrophages and promote pathogen clearance [76].

However, it should be noted that there are reports indicating that there was no beneficial effect of influenza vaccination on the risk of in-hospital mortality, indicating that vaccination did not change COVID-19-related morbidity and mortality [18]. Consistent with Conlon et al. (2021), who indicated that influenza-vaccinated patients were more likely to have a shorter hospital stay, in our study, the days of hospital stay were reduced in patients with influenza vaccination status [56]. Adjusting the analysis of this study, we observed that COVID-19-related deaths were not related to influenza vaccination status in patients, but were related to T2DM, heart disease, obesity, and smoking. Consistent with Conlon et al. (2021), who indicated that influenza-vaccinated patients were more likely to have a shorter hospital stay, in our study, the days of hospital stay were edited in patients with influenza vaccination status [56]. Adjusting the analysis of this study, we observed that COVID-19-related deaths were not related to influenza vaccination status in patients, but were related to T2DM, heart disease, obesity, and smoking. Pedote et al. (2021) also report finding no association between influenza vaccination and risk of death in patients with COVID-19; they also report that chronic diseases are the main risk factors for COVID-19-related hospitalization and death. 

In this study, the comorbidities associated with death caused by COVID-19 complications are T2DM, heart disease, and obesity [77,78]. A study in Spain showed that mortality in patients with COVID-19 was associated with age and comorbidities according to the Charlson comorbidity index, and not vaccination status; they consider comorbidities to be significant independent predictors of mortality [79]. In this study, the comorbidities associated with death caused by COVID-19 complications were T2DM, heart disease, and obesity [77,78]. Mortality in patients with COVID-19 was associated with age and comorbidities according to the Charlson comorbidity index, and not vaccination status [79]; comorbidities were independent predictors of mortality [79]. 

In this investigation we report the influence of influenza vaccination on COVID-19, which allowed us to estimate the association between influenza vaccination and its relationship with the presence of symptoms, evolution, and outcome of patients with COVID-19. Although influenza vaccination has no impact on severe manifestations of COVID-19, it should be promoted as an essential public health preventive measure throughout the world’s population because, as Paget et al. (2020) point out, avoiding influenza infections can facilitate the differential diagnosis of COVID-19 and free up healthcare facilities to care for these patients [80]. In addition, influenza vaccination is critical in protecting older adults, with higher vaccination coverage among the elderly, healthcare workers, vulnerable people with chronic diseases, and workers in other essential services. It can also influence the health effects and social consequences of the COVID-19 pandemic [14]. 

## 5. Conclusions

The findings of this study suggest that influenza vaccination was associated with a decrease in the number of symptoms in patients hospitalized due to COVID-19, as well as a decrease in the number of patients hospitalized with comorbidities. The duration of hospitalization in vaccinated patients was less and the survival of these patients was significantly higher than that of unvaccinated patients. While influenza vaccination did not correlate with the outcome in patients with COVID-19, comorbidities such as T2DM, heart disease, obesity, and tobaccoism increased the risk of death due to COVID-19. These results can guide clinicians for proper prognostication and treatment planning of patients.

## Figures and Tables

**Figure 1 vaccines-12-00765-f001:**
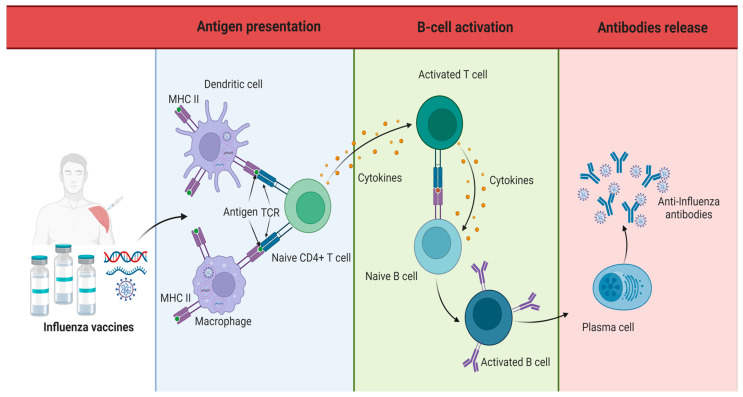
Production of anti-influenza antibodies. The generation of anti-influenza antibodies is generated prophylactically through the application of the influenza vaccine. Antibody formation begins with the presentation of antigens (components of the influenza virus) by dendritic cells and macrophages to naive CD4 lymphocytes, thereby generating activation of B and T lymphocytes. The activated B lymphocytes begin to produce antibodies on their surface, which undergo a maturation process, until finally, the plasma cells generate large quantities of mature antibodies specific for the identification and elimination of the influenza virus [21].

**Figure 2 vaccines-12-00765-f002:**
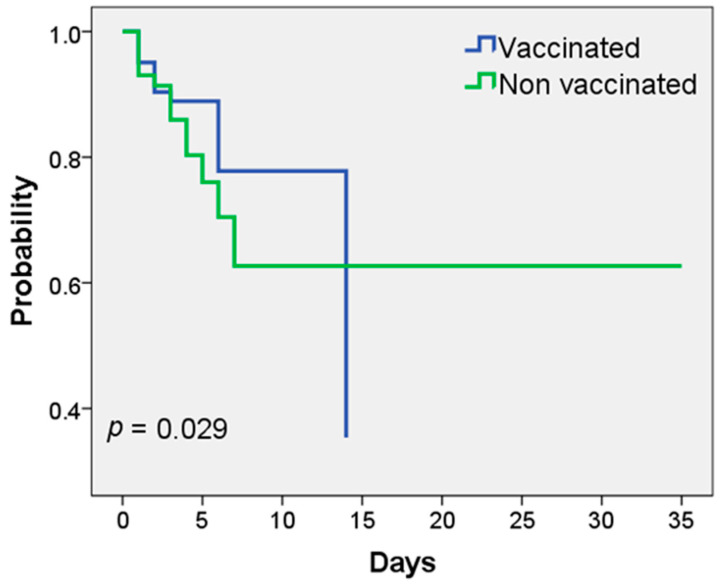
Survival analysis. Estimated Kaplan–Meier survival curve by influenza vaccination status (vaccinated patients: 509; unvaccinated patients: 1203) and days of hospitalization (vaccinated patients: 3.7 days on average; unvaccinated patients: 6.9 days on average) by COVID-19.

**Table 1 vaccines-12-00765-t001:** General characteristics of study population. Table displays the comparison of clinicopathological findings in hospitalized patients with COVID-19 with and without influenza vaccine.

Variables	Population with Influenza Vaccine *n* = 509 (%)	Population without Influenza Vaccine *n* = 1203 (%)	Odds Ratio (95% CI)	*p*-Value
Sex			1.2 (0.7–2.0)	0.05
M	309 (60.7)	669 (55.6)		
F	200 (39.2)	534 (44.3)		
Intubated	36 (7)	131 (10.8)	0.6 (0.4–0.9)	0.013
Fever	339 (66.6)	929 (77.2)	0.5 (0.4–0.7)	<0.001
Cough	509 (100)	1022 (84.9)	1.7 (1.1–2.7)	<0.001
Odymphagia	226 (44.4)	494 (41)	1.1 (0.9–1.4)	0.201
Irritability	28 (5.5)	72 (5.9)	0.9 (0.5–1.4)	0.696
Diarrhea	56 (11)	252 (20.9)	0.4 (0.3–0.6)	<0.001
Thoracic Pain	397 (77.9)	778 (64.6)	1.9 (1.5–2.4)	<0.001
Chills	283 (55.5)	528 (43.8)	1.6 (1.2–1.9)	<0.001
Headache	425 (83.4)	931 (77.3)	1.4 (1.1–1.9)	0.004
Myalgia	282 (55.4)	771 (64)	0.6 (0.5–0.8)	0.001
Arthralgia	282 (55.4)	771 (64)	0.6 (0.5–0.8)	0.001
Attack to the general state	453 (88.9)	1002 (83.2)	1.6 (1.1–2.2)	0.003
Rhinorrhea	86 (16.8)	290 (24.1)	0.6 (0.4–0.8)	0.001
Polypnea	256 (50.2)	810 (67.3)	0.4 (0.3–0.6)	<0.001
Vomiting	56 (11)	108 (8.9)	1.2 (0.8–1.7)	0.193
Abdominal Pain	84 (16.5)	184 (15.2)	1.0 (0.8–1.4)	0.530
Cyanosis	115 (22.5)	380 (31.5)	0.6 (0.4–0.8)	<0.001
Anosmia	140 (27.5)	478 (39.7)	0.5 (0.4–0.7)	<0.001
Dysgeusia	112 (22)	462 (38.4)	0.4 (0.3–0.5)	<0.001
T2DM *	197 (38.7)	594 (49.3)	0.6 (0.5–0.8)	<0.001
COPD ^†^	28 (5.5)	126 (10.4)	0.4 (0.3–0.7)	0.001
Asthma	29 (5.6)	48 (3.9)	1.4 (0.9–2.3)	0.119
Immunosuppression	58 (11.3)	110 (9.1)	1.2 (0.9–1.7)	0.152
Arterial Hypertension	283 (55.5)	757 (62.9)	0.7 (0.5–0.9)	0.005
Heart Disease	28 (5.5)	72 (5.9)	0.9 (0.5–1.4)	0.696
Obesity	285 (55.9)	615 (51.1)	1.2 (0.9–1.4)	0.065
Tobaccoism	56 (11)	169 (14)	0.7 (0.5–1.0)	0.088
Antiviral Treatment	28 (5.5)	92 (7.6)	0.7 (0.4–1.0)	0.112

* T2DM: type 2 diabetes mellitus; ^†^ COPD: chronic obstructive pulmonary disease.

**Table 2 vaccines-12-00765-t002:** Multivariate analysis. Multivariate logistic regression analysis to evaluate the relationships between variables associated with the outcome of COVID-19 in hospitalized patients.

Variable	Outcome		Crude Odds Ratio (95% CI)	*p*-Value	Adjusted Odds Ratio (95% CI)	*p*-Value
	Death *n* = 582 (%)	Survivors *n* = 1130 (%)				
Vaccinated	196 (33.6)	291 (25.7)	1.5 (1.1–1.8)	0.001	0.9 (0.6–1.2)	0.657
Non Vaccinated	386 (66.3)	839 (74.2)				
Intubated	139 (23.8)	28 (2.4)	1.0 (0.6–1.5)	0.01	12.8 (7.9–20.5)	<0.001
T2DM *	281 (48.2)	421 (37.2)	1.8 (1.3–2.3)	<0.001	2.3 (1.7–3)	<0.001
COPD ^†^	59 (10.1)	49 (4.3)	1.6 (0.1–2.6)	0.074	1.4 (0.9–2.4)	0.109
Asthma	30 (5.1)	43 (3.8)	0.6 (0.3–0.9)	0.128	0.6 (0.3–1.1)	0.149
Immunosuppression	62 (10.6)	47 (4.1)	0.9 (0.6–1.4)	0.706	1.2 (0.7–2.1)	0.366
Arterial Hypertension	332 (57)	613 (54.2)	1.1 (0.8–1.4)	0.002	0.7 (0.6–1.0)	0.097
Heart Disease	47 (8)	20 (1.7)	1.2 (1.1–1.2)	0.03	5.1 (2.6–9.7)	<0.001
Obesity	302 (51.8)	122 (10.7)	1.6 (1.2–2.1)	<0.001	9.5 (7–12.9)	<0.001
Tobaccoism	83 (14.2)	73 (6.4)	1.2 (0.6–1.9)	0.001	2.4 (1.6–3.8)	<0.001

* T2DM: type 2 diabetes mellitus. ^†^ COPD: chronic obstructive pulmonary disease.

## Data Availability

The data that support the findings of this study are available on request from the corresponding authors [M.L.M.-F. and L.A.-C.].

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
