# Peer review of "Evaluation of the Effect of Influenza Vaccine on the Development of Symptoms in SARS-CoV-2 Infection and Outcome in Patients Hospitalized due to COVID-19"

_vaccines, 2024, doi:10.3390/vaccines12070765_

Round 1
Reviewer 1 Report
Comments and Suggestions for Authors
The study is important in order to understand relationship between influenza vaccination and COVID-19. Unfortunately study design raises some questions.
1. COVID patients included in this study were admitted to the hospital during nearly one year period. There is no information when they were vaccinated against influenza relatively to admission. It is not clear what time was between, probably authors could make different groups. This is important because at least two mechanisms may be applied. First - overall activation of innate immunity if influenza vaccination was very close to COVID-19 disease onset. Second - activation of trained immunity mechanisms if vaccination was several months before COVID-19. Authors did not look for these possible differences and did not discuss it.
2. Age of studied patients was 20-86 years. Better to make 2-3 age groups because COVID-19 symptoms are more severe in elderly and because there is great influence of comorbidity in elderly.
3. Table 2 does not give information on comorbidity incidence depending on influenza vaccination. Number of survivors is much bigger in non vaccinated group. It does not meet with the paper conclusions.
So it is better to add some data to improve this paper.
Reviewer 2 Report
Comments and Suggestions for Authors
It is clear that such a global problem as a new pandemic requires a quick search for effective methods of protection. At first glance, it seems very appealing to think about the cross-protection immunity after flu vaccination of people who have COVID-19. However, immune protection after vaccination is based on the production of antibodies and the activation of cellular immune functions. These aspects are not mentioned in the article. The authors used statistical data processing only. Although there are many factors that determine the attitude of different groups of the population to their health. Adherence to vaccination with influenza vaccines is principally greater in more conscious and more financially well-off people. The milder course of COVID-19 in flu-vaccinated patients could be due to earlier admission, earlier hospitalization and better health of patients due to a better economic situation. It may be also other factors not mentioned in the article.
Advice to authors: since it is not possible to add the study of the level of antibodies for influenza viruses and relate it to the morbidity and mortality associated with COVID-19, it is proposed to study such factor as - number of the days from the onset of the COVID-19 to hospitalization day and influence this factor to the severe course of COVID-19 and disease outcome.
Round 2
Reviewer 1 Report
Comments and Suggestions for Authors
I have read authors response and especially discussion concerning answers to my review. I think now after improvement that was performed by authors this paper can be published.
Reviewer 2 Report
Comments and Suggestions for Authors
Everything is OK